# Transient Dynamics in the Random Growth and Reset Model

**DOI:** 10.3390/e23030306

**Published:** 2021-03-05

**Authors:** Tamás S. Biró, Lehel Csillag, Zoltán Néda

**Affiliations:** 1Wigner Research Centre for Physics, 1121 Budapest, Hungary; biro.tamas@wigner.hu; 2Complexity Science Hub, 1080 Vienna, Austria; 3Department of Physics, Babeş-Bolyai University, 400084 Cluj-Napoca, Romania; lehel@csillag.ro

**Keywords:** growth and reset process, master equation, stationary distribution, transient dynamics

## Abstract

A mean-field type model with random growth and reset terms is considered. The stationary distributions resulting from the corresponding master equation are relatively easy to obtain; however, for practical applications one also needs to know the convergence to stationarity. The present work contributes to this direction, studying the transient dynamics in the discrete version of the model by two different approaches. The first method is based on mathematical induction by the recursive integration of the coupled differential equations for the discrete states. The second method transforms the coupled ordinary differential equation system into a partial differential equation for the generating function. We derive analytical results for some important, practically interesting cases and discuss the obtained results for the transient dynamics.

## 1. Introduction

A challenge for physics in taming complexity is to create simple analytically treatable models, with a wide range of applicability [1]. Mean-field type master equations [2] with state-dependent transition rates represent such an example. Beside the diffusion and growth processes such equations were used with success in understanding distributions encountered in social or economic problems [3]. Recently Biró and Néda proposed an analytically solvable master equation with a unidirectional growth and a reset term [4]. For simple growth and reset rates the equation leads to stationary distributions which appear in many complex systems (for a review please consult [5]). Recent applications are simple but powerful theories for explaining scientific citation and Facebook share dynamics [6] or income distributions in modern societies [7]. It has been also proved that the proposed system converges to these stationary distributions independently of the initial condition [8]. For modeling purposes, it remains however an open question as to how fast the convergence to the stationary solution is. Comparison of distributions generated by the model with those derived from experimental data might be strongly biased if stationarity is not reached. Frequently the relaxations are exponential in time, and the largest exponent governs this convergence. In the present work, we intend to find an explicit formula for the law of convergence in some simple, analytically treatable cases. Instead of the continuous version of the growth and reset model we consider here the discrete version, where the allowed states of the system are taken from a discrete, but infinite set—similarly to the study performed in [9].

The present paper is organized as follows: first we present the simple growth and reset model for discrete states and its general stationary solution. We consider two specific solutions in compact forms: constant reset rate with constant growth rate and constant reset rate with linear growth rate (preferential growth). We then present two methods for studying the transient behavior: the recursive induction method and the generating function approach. Explicit solutions for some practically important cases are given in these two particular cases. Finally, we discuss the obtained results and exemplify numerically the unveiled convergence.

## 2. Master Equation for Unidirectional Growth with Reset

A general linear Master-equation for a system possessing discrete states, *n* writes as
(1)$$\frac{\partial P_{n}\left(t\right)}{\partial t}\equiv {\dot{P}}_{n}\left(t\right)=\sum_{m}\left[w_{nm}P_{m}\left(t\right)-w_{mn}P_{n}\left(t\right)\right],$$
where $P_{n}$ is the probability of the system being in state *n* and $w_{nm}$ are the state-dependent transition rates from the state *m* to *n*. In most of the cases, local transition rates are considered, such as for the classical diffusion with a drift [10]:(2)$$w_{nm}=\mu_{m}\delta_{n,m+1}+\lambda_{m}\delta_{n,m-1}.$$

This choice leads to the evolution equation
(3)$${\dot{P}}_{n}\left(t\right)=\mu_{n-1}P_{n-1}\left(t\right)+\lambda_{n+1}P_{n+1}\left(t\right)-\mu_{n}P_{n}\left(t\right)-\lambda_{n}P_{n}\left(t\right),$$
being the discrete model of one-dimensional diffusion, with position dependent drift and diffusion coefficients.

Here we concentrate on a different process: the local transition is allowed only in one direction, increasing *n* by one unit. As it is discussed in [5] such a process is relevant for many physical, biological and socio-economic phenomena. To make a stationary state possible, this local growth is supplemented with a non-local transition, which resets the system from any state to the ground state (n=0). Accordingly, we have thus
(4)$$w_{nm}=\mu_{m}\delta_{n,m+1}+\gamma_{m}\delta_{n,0}\;\;,$$
leading to the evolution equation
(5)$${\dot{P}}_{n}\left(t\right)=\mu_{n-1}P_{n-1}\left(t\right)+\delta_{n,0}\langle \gamma \rangle -\left(\mu_{n}+\gamma_{n}\right)P_{n}\left(t\right),$$
where $\langle \gamma \rangle =\sum_{j}\gamma_{j}P_{j}\left(t\right)$. Imposing the conservation of probability $\sum_{j}P_{j}\left(t\right)=1$, the above equation can be written as: (6)$$\begin{array}{c} {\dot{P}}_{n}\left(t\right)=\mu_{n-1}P_{n-1}\left(t\right)-\left(\mu_{n}+\gamma_{n}\right)P_{n}\left(t\right),\;n>0 \end{array}$$(7)$$\begin{array}{c} {\dot{P}}_{0}\left(t\right)=-\left(\mu_{0}+\gamma_{0}\right)P_{0}\left(t\right)+\langle \gamma \rangle ,\;n=0. \end{array}$$

This model is schematically illustrated in Figure 1.

## 3. Stationary Solution

For n>0 the stationary distribution $Q_{n}=\lim_{t\to \infty }P_{n}\left(t\right)$, is given by the condition:(8)$$\mu_{n-1}Q_{n-1}=\left(\mu_{n}+\gamma_{n}\right)Q_{n}.$$

The recursive solution of the above gives:(9)$$Q_{n}\;=\;\frac{\mu_{0}Q_{0}}{\mu_{n}}\prod_{j=1}^{n}\frac{\mu_{j}}{\mu_{j}+\gamma_{j}}\;=\;\frac{\mu_{0}Q_{0}}{\mu_{n}}\prod_{j=1}^{n}{\left(1+\frac{\gamma_{j}}{\mu_{j}}\right)}^{-1}.$$

For n=0 one obtains $Q_{0}$ from the normalization condition: $\sum_{i}^{\;}Q_{i}=1$. We consider in the following two simple cases, where the stationary distributions are given in compact form.

### 3.1. Constant Growth and Reset Rates

The simplest case is when both the growth rate $\mu_{n}=\mu$ and the reset rate $\gamma_{n}=\gamma$ are constant, i.e., independent of the initial and final states in the microtransition. In such cases the stationary distribution becomes an n-th power
(10)$$Q_{n}=Q_{0}{\left(1+\frac{\gamma }{\mu }\right)}^{-n},$$
with:(11)$$Q_{0}=1-\frac{1}{1+\frac{\gamma }{\mu }}=\frac{\gamma }{\mu +\gamma }.$$

This is the well known geometrical distribution and it can easily be transformed into an exponential law, as follows:(12)$$Q_{n}=\frac{\gamma }{\mu +\gamma }e^{-n\ln\left(1+\frac{\gamma }{\mu }\right)}.$$

If such a growth and reset process with constant growth and reset rates is relevant to a physical system, and the index *n* quantifies the energy of the state (i.e., $E_{n}=n\;\epsilon$), the equilibrium distribution reminds us to the famous Gibbs-Boltzmann distribution. The system behaves like a thermodynamic system in canonical ensemble with the “temperature”:(13)$$k_{B}T=\frac{\epsilon }{\ln\left(1+\frac{\gamma }{\mu }\right)}.$$

For such non-thermal applications, one can interpret this quantity as a generalized temperature. In special cases when the reset rate is much smaller than the growth rate γ≪μ, one obtains:(14)$$k_{B}T\;\approx \;\epsilon \;\frac{\mu }{\gamma }.$$

It worth mentioning here that the temperature-like combination for the growth and reset rate parameters identified through the stationary distribution is a particular appearance of the legendary “fluctuation-dissipation” theorem. When the rates are constant, their sets the temperature.

### 3.2. Linear Growth Rate and Constant Reset Rate

More interesting, and application-wise much more exciting is to consider a linear preference in the growth rate: $\mu_{n}=\sigma \left(n+1\right)$ and keeping the reset rate constant $\gamma_{n}=\gamma$. This is the mathematical realization of Matthew principle “For whosoever hath, to him shall be given, …”. Such effects are common in complex systems, for a review one can consult [11]. Here the state *n* can be thought of possessing *n* units of some arbitrary goods such as money, energy, scientific citations, etc.

Using Equation (9) and the linear preference condition we obtain:(15)$$Q_{n}=\frac{\sigma Q_{0}}{\sigma (n+1)}\prod_{j=1}^{n}\frac{\sigma (j+1)}{\sigma (j+1)+\gamma }=Q_{0}\frac{n!}{\prod_{j=1}^{n}\left(j+1+\frac{\gamma }{\sigma }\right)}.$$

Again, $Q_{0}$ can be calculated from the normalization condition $\sum_{n=0}^{\infty }Q_{n}=1$ and the normalized distribution becomes:(16)$$Q_{n}=\frac{\gamma }{\sigma +\gamma }\frac{n!}{\prod_{j=1}^{n}\left(j+1+\frac{\gamma }{\sigma }\right)}=\frac{\gamma }{\sigma +\gamma }\frac{\Gamma \left(n+1\right)\;\Gamma \left(2+\frac{\gamma }{\sigma }\right)}{\Gamma \left(n+2+\frac{\gamma }{\sigma }\right)}\;.$$

This is the Waring distribution [12], familiar from failure statistics. In the limit n→∞ it leads to the asymptotic behavior:(17)$$Q_{n\to \infty }\approx \frac{\gamma }{\left(\sigma +\gamma \right)}\;\Gamma \left(2+\frac{\gamma }{\sigma }\right)\;n^{-1-\frac{\gamma }{\sigma }}.$$

Whenever the reset is rare, i.e., γ≪σ, we arrive at the well-known Zipf distribution [13], which is frequently encountered in many complex systems [14,15]:(18)$$Q_{n\to \infty }∼n^{-1}.$$

## 4. Convergence towards Stationarity for Constant Reset and Growth Rates

For constant reset and growth rates ($\gamma_{n}=\gamma$, $\mu_{n}=\mu$) we used two different methods for studying the transient dynamics in the growth and reset model. Both of them leads to the same compact analytical solution.

### 4.1. The Recursive Substitution Method

We rewrite the system of Equations (6) and (7) in a matrix form:(19)$$\dot{P}=\left[\begin{array}{ccccc} -(\gamma +\mu ) & 0 & 0 & 0 & 0\\ \mu & -(\gamma +\mu ) & 0 & 0 & 0\\ 0 & \mu & -(\gamma +\mu ) & 0 & 0\\ 0 & 0 & \ddots & \ddots & -(\gamma +\mu ) \end{array}\right]P+\left[\begin{array}{c} \langle \gamma \rangle \\ 0\\ 0\\ 0\\ \vdots \end{array}\right].$$

When γ is constant, its expectation value, 〈γ〉, does not depend on time and the system can be handled by solving recursively the equations, repeatedly substituting the obtained solution for *n* into the equation for n+1. We arrive at the transient behavior
(20)$$P_{n}\left(t\right)=e^{-(\gamma +\mu )t}\sum_{i=0}^{n}C_{i}\frac{t^{n-i}}{(n-i)!}\mu^{n-i}+\frac{\gamma }{\mu }\frac{\mu^{n+1}}{{\left(\mu +\gamma \right)}^{n+1}},$$
where the $C_{i}$-s are integration constants that are determined from the $P_{n}\left(0\right)$ initial conditions. For t→∞ the first term will vanish because the exponential term dominates the polynomial, therefore we get:(21)$$Q_{n}=\lim_{t\overset{}{\to }\infty }P_{n}=\frac{\gamma }{\mu }\frac{\mu^{n+1}}{{\left(\mu +\gamma \right)}^{n+1}}=\frac{\gamma }{\mu (1+\frac{\gamma }{\mu })}\frac{1}{{\left(1+\frac{\gamma }{\mu }\right)}^{n}}=\frac{\gamma }{\gamma +\mu }{\left(1+\frac{\gamma }{\mu }\right)}^{-n}.$$

This is exactly the result announced in (10) and (11) demonstrating that the system converges to the presumed stationary distribution, in agreement with the classical H-theorem for master-equations [16].

We consider now a specific initial condition:(22)$$P_{0}\left(0\right)=1,$$
(23)$$P_{n}\left(0\right)=0,\;n>0.$$

In this case we obtain
(24)$$C_{0}=1-\frac{\gamma }{\gamma +\mu }=1-\frac{\gamma \mu^{0}}{\gamma +\mu },$$
(25)$$C_{n}=-\frac{\gamma \mu^{n}}{{\left(\gamma +\mu \right)}^{n+1}},$$
leading to the compact form:(26)$$P_{n}\left(t\right)=e^{-t(\gamma +\mu )}\frac{{\left(t\mu \right)}^{n}}{n!}+\frac{\mu^{n}\gamma }{{\left(\gamma +\mu \right)}^{n+1}}\left[1-\frac{\Gamma \left[n+1,t(\gamma +\mu )\right]}{n!}\right].$$

Using the properties of the Γ[q,x] upper incomplete gamma function in the limit t→∞
(27)$$\Gamma \left[q,x\to \infty \right]\approx x^{q-1}\;e^{-x},$$
we get:(28)$$P_{n}\left(t\to \infty \right)\approx \frac{\mu^{n}\gamma }{{\left(\gamma +\mu \right)}^{n+1}}+\frac{{\left(t\mu \right)}^{n}}{n!}e^{-t(\gamma +\mu )}\frac{\gamma }{\gamma +\mu }.$$

In agreement with the intuitive reasoning, this result suggests that the convergence is quick if γ is large and μ is small.

An important and simply treatable case is when t(γ+μ)→0. In this case we use the property Γ[n+1,0]=n! leading in this limit to:(29)$$P_{n}\left(t\right)∼\frac{{\left(t\mu \right)}^{n}}{n!}.$$

This result indicates that at the beginning of the dynamics one observes a Poisson distribution instead of the exponential one.

### 4.2. Generating Function Method

The generating function of the $P_{n}\left(t\right)$ distribution writes as:(30)$$G\left(t,z\right)=\sum_{n=0}^{\infty }P_{n}\left(t\right)e^{-nz}.$$

Normalization of $P_{n}\left(t\right)$ leads to:(31)G(t,0)=1.

The initial conditions $P_{n}\left(0\right)$ defines
(32)$$G\left(0,z\right)=\sum_{n=0}^{\infty }P_{n}\left(0\right)e^{-nz}=G_{0}\left(z\right),$$
and the stationary state defines:(33)$$G\left(\infty ,z\right)=\sum_{n=0}^{\infty }Q_{n}e^{-nz}=G_{\infty }\left(z\right).$$

The deviation from stationarity at time *t* is
(34)$$\Delta_{n}\left(t\right)=P_{n}\left(t\right)-Q_{n},$$
and in the generating function formalism it takes the following form:(35)$$\Delta \left(t,z\right)=G\left(t,z\right)-G_{\infty }\left(z\right).$$

For constant rates ($\gamma_{n}=\gamma$ and $\mu_{n}=\mu$), the generating function at the stationary state can be easily computed:(36)$$G_{\infty }\left(z\right)=\frac{\frac{\gamma }{\mu }\;e^{z}}{\left(1+\frac{\gamma }{\mu }\right)\;e^{z}-1}.$$

We can now rewrite the growth and reset master Equation (6) in the *G* space, leading to:(37)$$\frac{\partial G(t,z)}{\partial t}=\mu \;G\left(t,z\right)e^{-z}-\left(\mu +\gamma \right)\;G\left(t,z\right)+\gamma .$$

The above equation can be integrated and gives the solution
(38)$$G\left(t,z\right)=c\left(z\right)\;e^{\mu \;t\;\left(e^{-z}-1-\frac{\gamma }{\mu }\right)}+\frac{\frac{\gamma }{\mu }}{1+\frac{\gamma }{\mu }-e^{-z}}=c\left(z\right)\;e^{\mu \;t\;\left(e^{-z}-1-\frac{\gamma }{\mu }\right)}+G_{\infty }\left(z\right),$$
with c(z) a function that can be determined from the initial conditions. The convergence to the stationary state in the generating function formalism is immediate:(39)$$G\left(\infty ,z\right)=\lim_{t\to \infty }\left(c\left(z\right)\;e^{\mu \;t\;\left(e^{-z}-1-\frac{\gamma }{\mu }\right)}+\frac{\frac{\gamma }{\mu }}{1+\frac{\gamma }{\mu }-e^{-z}}\right)=\frac{e^{z}\frac{\gamma }{\mu }}{\left(1+\frac{\gamma }{\mu }\right)e^{z}-1}.$$

From Equation (38) and using c(z)=Δ(0,z) we arrive at
(40)$$\Delta \left(t,z\right)=e^{-\gamma t}e^{\mu t(e^{-z}-1)}\Delta \left(0,z\right).$$

In the above solution one observes the generating function of the Poisson-distribution. By straightforward mathematical steps that are detailed in the Appendix A, we arrive at:(41)$$\Delta_{n}\left(t\right)=e^{-(\gamma +\mu )t}\sum_{k=0}^{n}\frac{{\left(\mu t\right)}^{k}}{k!}\Delta_{n-k}\left(0\right).$$

Using the definition (35), the initial conditions (22), (23) and the form of the stationary distribution given by (12) we get the same result as the one given in (26).

## 5. Constant Reset Rate and Linearly Increasing Growth Rate

### 5.1. The Recursive Substitution Method

We proceed in a similar manner as before and rewrite the discrete evolution equation into a matrix form. We consider for the moment arbitrary state dependent growth rates $\mu_{n}$ and a constant reset rate, γ:(42)$$\dot{P}=\left[\begin{array}{ccccc} -(\mu_{0}+\gamma ) & 0 & 0 & 0 & 0\\ \mu_{0} & -(\mu_{1}+\gamma ) & 0 & 0 & 0\\ 0 & \mu_{1} & -(\mu_{2}+\gamma ) & 0 & 0\\ 0 & 0 & \ddots & \ddots & -(\mu_{n}+\gamma ) \end{array}\right]P+\left[\begin{array}{c} \langle \gamma \rangle \\ 0\\ 0\\ 0\\ \vdots \end{array}\right].$$

Proceeding now with the calculations as done for the constant growth rate, assuming no degeneracy in the values of $\left(\mu_{i}+\gamma \right)$ (i.e., all $\mu_{i}$ values are different) we achieve the following result by mathematical induction:(43)$$P_{n}=\sum_{k=0}^{n}C_{k}e^{-(\mu_{k}+\gamma )t}\frac{\prod_{i=k}^{n-1}\mu_{i}}{\prod_{j=k+1}^{n}\left(\mu_{j}-\mu_{k}\right)}+\frac{\gamma }{\mu_{n}}\prod_{i=0}^{n}\frac{\mu_{i}}{\mu_{i}+\gamma }\;,$$
where $C_{i}$ are integration constants that will be determined from the initial conditions. We also assumed in the product notation $\prod_{m}^{n}\left(\ldots \right)=1$ if m>n by definition.

Due to the fact that in the first term the exponential has a negative exponent, in the t→∞ limit it vanishes, reconstructing the stationary distributions (9):(44)$$Q_{n}=\lim_{t\overset{}{\to }\infty }P_{n}=\frac{\gamma }{\mu_{n}}\prod_{j=0}^{n}\frac{\mu_{j}}{\mu_{j}+\gamma_{j}}\;.$$

By fixing the initial conditions $P_{n}\left(0\right)$ it is possible to determine the integration constants, $C_{k}$, in a recursive manner by solving a lower triangular system of equations. Introducing the notations
(45)$$\alpha_{k}^{n}=\frac{\prod_{i=k}^{n-1}\mu_{i}}{\prod_{j=k+1}^{n}\left(\mu_{j}-\mu_{k}\right)}\;,$$
(46)$$\beta_{n}=\frac{\gamma }{\mu_{n}}\prod_{i=0}^{n}\frac{\mu_{i}}{\mu_{i}+\gamma }\;,$$
we get the recursive form:(47)$$C_{n}=-\frac{1}{\alpha_{n}^{n}}\left(-P_{n}\left(0\right)+\beta_{n}+\sum_{k=0}^{n-1}C_{k}\alpha_{k}^{n}\right)\;.$$

In the case of linearly increasing growth rates, $\mu_{n}=\sigma \left(n+1\right)$, and for the initial conditions given in (22) and (23) one will get a complicated but still analytically computable form after substituting these values.

### 5.2. Generating Function Method

For constant reset rates but general growth rates the differential equation in the generator function formalism can be rewritten as an operator-equation. To do so we recall the obvious identity: (48)$$\sum_{i=1}^{\infty }f\left(n\right)P_{n}\left(t\right)e^{-nz}=\sum_{i=1}^{\infty }f\left(-\frac{\partial }{\partial z}\right)P_{n}\left(t\right)e^{-nz}=f\left(-\frac{\partial }{\partial z}\right)\sum_{i=1}^{\infty }P_{n}\left(t\right)e^{-nz}=f\left(-\frac{\partial }{\partial z}\right)G\left(t,z\right)\;.$$

Introducing thus the $\mu \left[-\frac{\partial }{\partial z}\right]$ operator corresponding to the $\mu_{n}$ growth rate (replacing in $\mu_{n}=\mu \left(n\right)$ the value of *n* by $-\frac{\partial }{\partial z}$), we get the following partial differential equation:(49)$$\frac{\partial \Delta }{\partial t}=\left(e^{-z}-1\right)\mu \left[-\frac{\partial }{\partial z}\right]\Delta -\gamma \Delta \;.$$

This evolution equation is analogous to the well-known Schrödinger equation. One can immediately write the solution as:(50)$$\Delta \left(t,z\right)=e^{t\left[\left(e^{-z}-1\right)\mu \left(-\frac{\partial }{\partial z}\right)-\gamma \right]}\Delta \left(0,z\right)\;.$$

We follow now the same path as for the constant growth rate case. Let us consider the growth rate in the form:(51)$$\mu_{n}=\sigma \left(n+1\right)\;.$$

The final stationary distribution in this case is given by:(52)$$G\left(\infty ,z\right)=\sum_{n=0}^{\infty }\frac{\gamma }{\sigma +\gamma }\frac{n!}{\prod_{j=1}^{n}\left(j+1+\frac{\gamma }{\sigma }\right)}e^{-nz}\;.$$

By introducing the notation
(53)$$\Phi \left(t,z\right)=\sigma t\left(1-e^{-z}\right)=\sigma t\Phi \left(z\right),$$
we get:(54)$$\Delta \left(t,z\right)=e^{-\gamma t}e^{\sigma \;t\;\Phi \left(z\right)\left(\frac{d}{dz}-1\right)}\Delta \left(0,z\right).$$

In the exponent there is a −1 shift, which we can treat as:(55)$$\Phi \left(z\right)\left(\frac{d}{dz}-1\right)f\left(z\right)=e^{z}\Phi \left(z\right)\frac{d}{dz}\left(e^{-z}f\left(z\right)\right)\;.$$

Using the $y=\int \frac{dz}{\Phi (z)}$ substitution:(56)$$\Phi \left(z\right){\left.\left(\frac{d}{dz}-1\right)f\left(z\right)\right|}_{z=z[y]}=e^{z}\Phi \left(z\right){\left.\frac{d}{dz}\left(e^{-z}f\left(z\right)\right)\right|}_{z=z[y]}=e^{z(y)}\frac{d}{dy}\left(e^{-z(y)}f\left(z\left[y\right]\right)\right)\;.$$

Considering the form of Φ(z) we get
(57)$$y=\int \frac{dz}{1-e^{-z}}=\int \frac{e^{z}}{e^{z}-1}dz=\log\left(e^{z}-1\right)+K\;,$$
from where it emerges the solution:(58)$$z\left[y\right]=\log\left(1+e^{y-K}\right)\;.$$

The property (56) is inherited to the *n*-th power
(59)$${\left.{\left[\Phi \left(z\right)\left(\frac{d}{dz}-1\right)\right]}^{n}f\left(z\right)\right|}_{z=z[y]}={\left[e^{z[y]}\frac{d}{dy}e^{-z[y]}\right]}^{n}f\left(z\left[y\right]\right)=e^{z[y]}\frac{d^{n}}{dy^{n}}\left(e^{-z[y]}f\left(z\left[y\right]\right)\right)\;,$$
and accordingly the solution of (54) can be written as:(60)$$\Delta \left(t,z\left[y\right]\right)=e^{-\gamma t}e^{z[y]}e^{\sigma t\frac{d}{dy}}\left(e^{-z[y]}\Delta \left(0,z\left[y\right]\right)\right)\;.$$

We recognize here the translational operator with σt, therefore:(61)$$\Delta \left(t,z\left[y\right]\right)={\left.e^{-\gamma t}e^{z[y]}\left(e^{-z}\Delta \left(0,z\right)\right)\right|}_{z=z[y+\sigma t]}.$$

Using now the definition of Δ(0,z) we get:(62)$$\Delta \left(t,z\left[y\right]\right)=e^{-\gamma t}e^{z[y]}\sum_{n=0}^{\infty }e^{-z[y+\sigma t](n+1)}\Delta_{n}\left(0\right)\;.$$

In the case of linearly increasing growth rate, Equations (57) and (58) will lead us to:(63)$$z\left[y+\sigma t\right]=\log\left(1+e^{y+\sigma t-K}\right)=z\left[y\right]+\sigma t+\log\left(1-e^{-z[y]}\left(1-e^{-\sigma t}\right)\right).$$

It follows therefore that:(64)$$e^{z[y]}e^{-(n+1)z[y+\sigma t]}=e^{-n\;z[y]}e^{-(n+1)\sigma t}\cdot {\left(1-e^{-z[y]}\left(1-e^{-\sigma t}\right)\right)}^{-(n+1)}\;.$$

With the help of the negative binomial expansion formula
(65)1−e−z[y](1−e−σt)−(n+1)=∑k=0∞n+kke−kz[y](1−e−σt)k,
we immediately get:(66)Δ(t,z[y])=e−γt∑n=0∞Δn(0)∑k=0∞e−nz[y]e−(n+1)σtn+kke−kz[y](1−e−σt)k.

Finally, we regroup the two sums as follows: we take a sum with respect to r=n+k and the remaining terms are contained in a sum with respect to *k*:(67)Δ(t,z[y])=e−(γ+σ)t∑r=0∞e−rz[y]∑k=0rΔr−k(0)(e−σt)r−k(1−e−σt)krk.

Using now the definition of Δ(t,z[y]) from Equation (35)
(68)∑n=0∞Δn(t)e−nz[y]=e−(γ+σ)t∑r=0∞e−rz[y]∑k=0rΔr−k(0)(e−σt)r−k(1−e−σt)krk,
and equating the corresponding terms leads to the solution we are looking for:(69)Δn(t)=e−(γ+σ)t∑k=0nnk(e−σt)n−k(1−e−σt)kΔn−k(0).

This is again a compact form which can be analytically used if one knows the $\Delta_{n}\left(0\right)=P_{n}\left(0\right)-Q_{n}$ initial conditions. As a particular case one can consider for example the $P_{n}\left(0\right)$ initial conditions given by (22) and (23).

## 6. Discussion on the Convergence Properties

From the compact formulas obtained with the generating function method it is easy to show also the convergence to stationarity.

### 6.1. Constant Growth and Reset Rate

In this case according to Equation (41) one can write:(70)$$\begin{array}{c} |\Delta_{n}\left(t\right)|\leq e^{-(\gamma +\mu )t}\sum_{k=0}^{n}\frac{{\left(\mu t\right)}^{k}}{k!}|\Delta_{n-k}\left(0\right)|<e^{-(\gamma +\mu )t}\sum_{k=0}^{\infty }\frac{{\left(\mu t\right)}^{k}}{k!}\left|\Delta_{n-k}\left(0\right)\right|\leq \\ \leq e^{-(\gamma +\mu )t}\sum_{k=0}^{\infty }\frac{{\left(\mu t\right)}^{k}}{k!}=e^{-(\gamma +\mu )t}e^{\mu t}=e^{-\gamma t}. \end{array}$$

Here we used that $|\Delta_{n-k}\left(0\right)|\leq 1$ according to the definitions of the probabilities. Since $\lim_{t\to \infty }e^{-\gamma t}=0$ the convergence is proven.

One can now investigate also the nature of convergence to the stationary distribution. We consider for this a particular but practically important case (considered also in Section 4.1), where the initial conditions are:(71)$$\begin{array}{c} P_{0}\left(0\right)=1,\\ P_{n}\left(0\right)=0,\;\;n>0\;. \end{array}$$

From Equations (10), (11) and (26) results:(72)$$\Delta_{n}\left(t\right)=P_{n}\left(t\right)-Q_{n}=e^{-t(\gamma +\mu )}\frac{{\left(t\mu \right)}^{n}}{n!}-\frac{\mu^{n}\gamma }{{\left(\gamma +\mu \right)}^{n+1}}\frac{\Gamma [n+1,t(\gamma +\mu )]}{n!}\;.$$

By using the derivative of the incomplete Gamma function
(73)$$\frac{\partial \Gamma [n,x]}{\partial x}=-x^{\left(n-1\right)}\;e^{-x},$$
after some tedious but straightforward algebra we arrive at
(74)$$\frac{d\Delta_{n}\left(t\right)}{dt}=\frac{\mu^{n}}{n!}e^{-(\gamma +\mu )\;t}\;t^{n-1}\;\left(n-\mu \;t\right).$$

For the initial conditions defined by Equation (71) we have $\Delta_{0}\left(0\right)>1$ and $\Delta_{n}\left(0\right)<0$ for n>0. For several *n* values the general shape of the time-evolutions of $\Delta_{n}\left(t\right)$ are plotted in Figure 2.

Apart from the n=0 case the convergence is non-monotonic. For the chosen initial conditions (71) and for n>0 at the beginning $\Delta_{n}\left(t\right)<0$ and $|\Delta_{n}\left(t\right)|$ decreases. After passing 0, it increases again and finally enters in the monotonic decreasing regime for $t>t_{c}$. The $t_{c}$ time-moment can be determined from the $\partial \Delta_{n}{\left(t\right)/\partial t|}_{t_{c}}=0$ condition. From Equation (74) it results: $t_{c}=n/\mu$. Turning around this condition, for a time moment *t* we can get critical state $n_{c}=$ Integer (μt), up to where the convergence is already monotonic. The above results are illustrated by plotting Equation (41) as a function of *n* for different time moments *t* parametrized by various γ and μ values (Figure 3). Apart from the obvious convergence to the stationary distribution, one also observes in the γ=1 and μ=2 case that $n_{c}$ increases with *t*.

### 6.2. Constant Reset Rate and Linearly Increasing Growth Rate

Using Equation (69) we obtain the time-dependent deviation from the stationary distribution as
(75)|Δn(t)|≤e−(γ+σ)t∑k=0nnk(e−σt)n−k(1−e−σt)k|Δn−k(0)|<<e−(γ+σ)t∑k=0nnk(e−σt)n−k(1−e−σt)k=e−(γ+σ)t.

Here we used again the fact that $|\Delta_{n-k}\left(0\right)|\leq 1$ and the binomial theorem to show that ∑k=0nnk(e−σt)n−k(1−e−σt)k=1. According to the above inequality we see that $\lim_{t\to \infty }\left|\Delta_{n}\left(t\right)\right|=0$.

In this case, Formula (69) for $\Delta_{n}\left(t\right)$ and also the $Q_{n}$ stationary distribution given by Equation (16) is more complicated than it was for constant γ and μ values. Even in the case of simple initial condition given by Equation (71) we couldn’t find a compact analytical formula that would allow some conclusions on the monotonic nature of the convergence. Similarly with the constant reset and growth rates case, it is possible however to visualize the convergence given by Equation (69). For the considered simple initial conditions (71) the general trend for the time evolution of the $\Delta_{n}\left(t\right)$ and $|\Delta_{n}\left(t\right)|$ values are rather similar to the case of the constant reset and growth rates and it is illustrated in Figure 4. According to this, for a fixed *n* value there is again a $t_{c}$ time-moment, so that for $t>t_{c}$ the convergence is monotonic. Similarly with the constant γ and μ case, the $t_{c}$ time-moment is increasing with the value of *n*.

For various γ and σ values we can visualize also for different time moments the $|\Delta_{n}\left(t\right)|$ values as a function of the state index *n*. Results in this sense are plotted in Figure 5. The figures illustrates nicely that there is an intermediate time moment $t_{n}$ where the $\Delta_{n}\left(t_{n}\right)|$ drops strongly due to the change from negative $\Delta_{n}$ to positive $\Delta_{n}$ values as it is illustrated in the general trend in the dynamics in Figure 4. We can also see that as *t* evolves, this minima shifts to higher *n* values.

## 7. Conclusions

We studied the convergence to the stationary distribution of the growth and reset master equation [5] applied to discrete states in two cases: (i) constant growth and reset rate, and (ii) constant reset rate with linearly increasing growth rate. Two different methods were used for obtaining analytical expressions describing this convergence. Our calculations can be viewed as an alternative proof of the convergence to stationarity and it differs from the one given by Biró, Néda and Telcs using the concept of generalized entropy [8]. The main advantage of the present method is that it not only proves that the stationary distributions are stable fix-points, but also gives a compact formula for the transient behavior of the relevant distribution functions. The revealed transient behavior is of importance, since in many complex systems we usually do not observe the stationary distributions experimentally, but only the distributions characteristic for the transient dynamics. For example, using our result for the case of constant reset and growth rates, Equation (26), we realize that one would fallaciously predict a Poisson distribution instead of the exponential one if studying the initial phase of the dynamics only. We obtained the explicit solution for the practically important $\mu_{n}=\sigma \left(n+1\right)$ and $\gamma_{n}=\gamma$ case, too. This is the case when the Matthew or “rich gets richer” principle holds and, as we discussed previously [5,6], it is relevant to a large number of social phenomena. Our results in this respect are interesting for the application of the growth and reset model when interpreting experimentally derived distributions in complex social systems, such as income or wealth distributions, citation statistics in science or even Facebook shares or YouTube likes on the Internet.

## Figures and Tables

**Figure 1 entropy-23-00306-f001:**
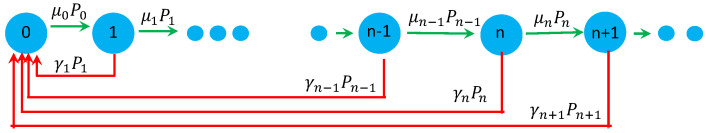
Schematic illustration of the unidirectional random growth with resetting.

**Figure 2 entropy-23-00306-f002:**
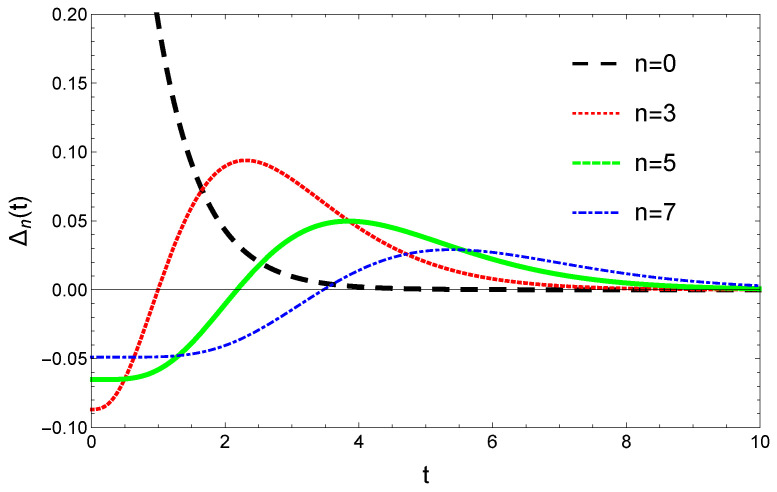
Time evolution and convergence of $\Delta_{n}\left(t\right)=P_{n}\left(t\right)-Q_{n}$ resulting from Equation (72) for γ=0.2, μ=1.3 and different *n* values.

**Figure 3 entropy-23-00306-f003:**
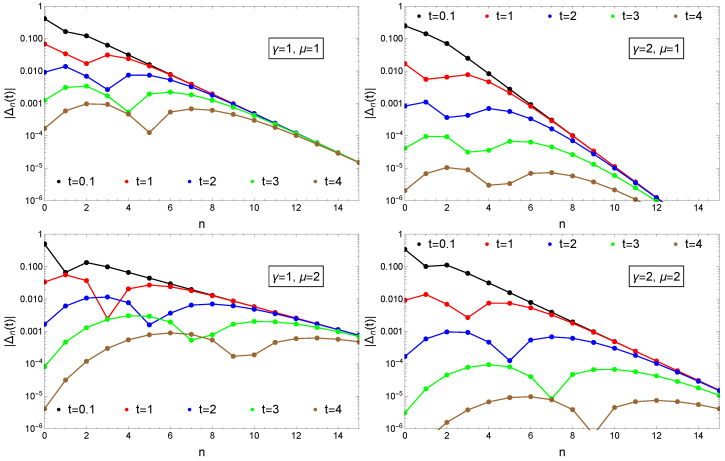
Convergence to the stationary distribution measured by $|\Delta_{n}\left(t\right)|=|P_{n}\left(t\right)-Q_{n}|$ in the case of constant γ reset rate and constant μ growth rates. Initial condition are given by Equation (71). Results are presented for different time-moments as indicated in the legend and for various γ and μ values as shown in the insets.

**Figure 4 entropy-23-00306-f004:**
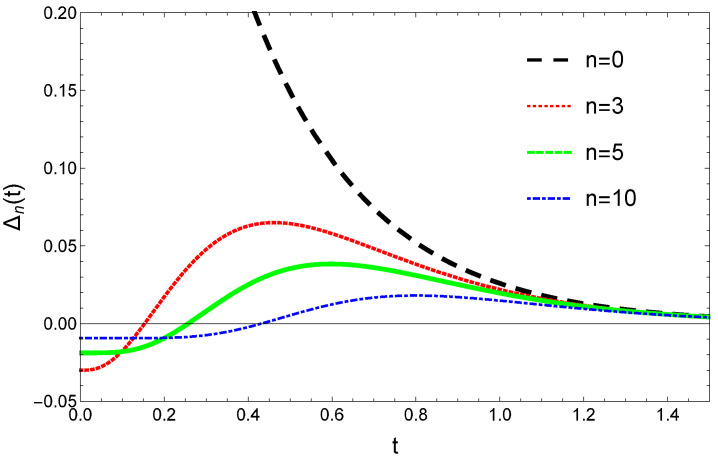
Time evolution and convergence of $\Delta_{n}\left(t\right)=P_{n}\left(t\right)-Q_{n}$ resulting from Equation (69) for γ=0.5 and σ=3 and different *n* values.

**Figure 5 entropy-23-00306-f005:**
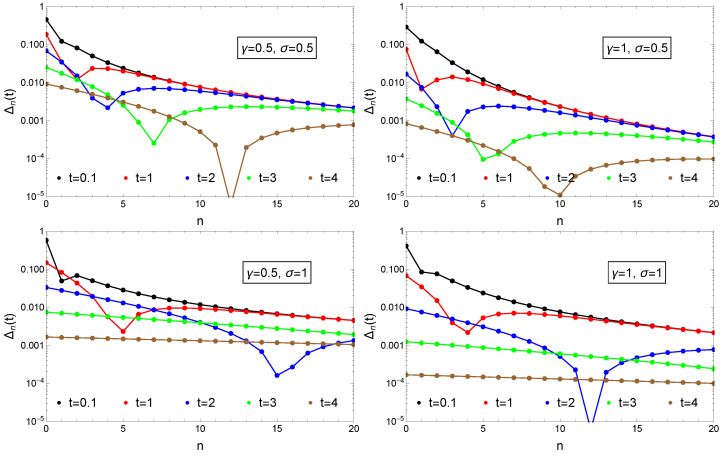
Convergence to the stationary distribution measured by $|\Delta_{n}\left(t\right)|=|P_{n}\left(t\right)-Q_{n}|$ for a constant reset rate, γ, and linearly increasing growth rates, μ=σ(n+1). Initial condition are given by Equation (71). Results are presented for different time-moments as indicated in the legend and for various γ and σ values as shown in the insets.

## Data Availability

Not applicable.

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
