# Peer review of "Transient Dynamics in the Random Growth and Reset Model"

_entropy, 2021, doi:10.3390/e23030306_

Round 1

Reviewer 1 Report

The mathematical problem of unidirectional random growth with reset is interesting and is also relevant to a number real-life complex socio-economic and financial systems.

The authors present a very clear description of the model and derive solutions for the stationary distribution and for and the transient dynamics.

While the mathematics based on the Master Equation formulation is somewhat involved, in particular for the transient dynamics, the authors do a great job presenting their results very clearly and in a well-organized fashion. The two methods the authors apply to the transient-dynamics problem (recursive and generating function, an alternative to generalized entropy) are rather interesting and yield insightful results, and because of their clear presentation, other researcher can greatly utilize this work as a go-to resource for these techniques (as they are better explained in-detail here than in most textbooks).

I recommend it for publication in Entropy, a highly suitable venue for this work.

Author Response

We thank the reviewer for his/her appreciation of our work.

Reviewer 2 Report

See attached file.

Author Response

Please consider the attached PDF file.

(below is the .TEX version)

===================================================

We acknowledged the comments and criticism formulated by the Reviewer. We respond here to the comments following the numbering from the report.   \\

{\bf Concerning the general impression of the Reviewer:} \\

The scarcity of physical comments is due to concentrating on technical issues; in a review (which was done in Reference 5.) indeed a more ample presentation of applications were in order. The present manuscript is not a review. It is a special article presenting the derivation of the time evolution of PDF-s in two particular, but also very common cases. Applications of the LGGR model to real world data and modeling challenges in many complex systems  have been published already earlier (Reference 5).\\

{\bf Detailed comments of the Reviewer}\\

1) The use of the term ''local'' means changing the state index only by one unit in the micro-process. Diffusion allows for both raising and lowering the index, here we single out the growth process. This is not a diffusion. As it is detailed in Reference 5 there are many non-physical processes where such evolution equation is relevant. We have 
added a small comment now. 

2) We made this part more clear now. In case when one deals with a physical system where the basic entities have discrete states that are characterized with energies that are dependent on the state index, $n$, and  if the growth and reset process with constant growth and reset rates is relevant to the time evolution of the system $\rightarrow$ the stationary distribution will be analogous  to the famous Gibbs-Boltzmann distribution.  The system behaves thus like a thermodynamic system in canonical ensemble with a  "temperature" that is governed by the growth ($\mu$) and reset $\gamma$ rates. 

3) Yes, we agree with the Referee and inserted a comment in such sense. Temperature like combination of growth and reset rate parameters identified through the stationary distribution does have physical implications: this is a particular appearance of the legendary "fluctuation-dissipation" theorem. When the rates are constant, their ratio is the temperature. Other cases generalize this correspondence.

4) The Zipf distribution appears in the limit $\gamma <<  \mu$. It is relevant, just not here but where it has been discovered, e.g. for occurrence frequencies of letters in literature text passages. Zipf PDF is stationary to such a dynamics in the above cited limit. Any other dynamics, also leading to a Zipf stationary PDF, must contain such a factorizable element. We gave two relevant references for the universal nature of Zipf's distribution. 

5) Yes, the usual H-theorem applies.  Instead of "proving" we should have written: "demonstrating". We have corrected the formulation and made connections to Boltzmann's H theorem. A new bibliographic  reference was added. 

6) We are not sure how general is the early times Poisson distribution in a general case. In the case of linear preference in the growth rate rather a Bernoulli distribution arises. Fallacious predictions based on too short observation times seem to be a general danger in data mining. Here we present a simple example where it looks trivial, nevertheless we found it worth mentioning. 

7) We may inflate the reference list with textbook and classical citations. Indeed the LGGR model has not been studied by many others - as far as we are aware of. Citations of works dealing with interesting research on exponential (geometric) and Waring distribution, etc. may add to the reference list; the problem is where to draw the border. Nevertheless  we include three other references as we detailed in the previous points.

8) Indeed, also based upon the other Referee report, we have moved one of the derivations to an Appendix section.\\

We are grateful for the Reviewer for his/her critical reading and suggestions. 

Reviewer 3 Report

This article presents explicit expressions for the time dependence of the probability distributions in two simple, though relevant, mean-field type models, and it is therefore of interest.
However, I have doubts concerning some technical points.
Moreover, the presentation - which is often careless - definitely needs improvement.

Technical points:

1. My main criticism concerns the discussions of monotonicity in sections 6.1 and 6.2, which seems to be invalid.
The point is that the values of $\Delta_i(0)$ are certainly not all positive, and therefore the absolute value
of $\Delta_n(t)$ is not guaranteed to be equal to the sum over the absolute values of the $\Delta_i(0)$.
More precisely, the first equality in eq. (75) is not in fact an equality, but an inequality, where the equality should be replaced with a less or equal $\leq$ sign. This renders the inequality (77) invalid, because there is no control over the denominator. This does not affect (75), since an inequality in place of the spurious equality still gives the result.
Nevertheless, this proof of simple convergence here is not relevant, since it was already proven in (20,21).
Similar considerations apply to the discussion in section 6.2.

2. It is unclear how and where Eq (28) is coming from, so the authors are urged to explain it.

3. Expression (57) seems to be incorrect. The product should be from j=1 to j=n-1, and in the denominator should be
$\lambda_n$, instead of $\gamma + \sigma$

4. I was unable to check Eq. (74). The authors are urged to check it, and expressions leading to it, carefully.

The authors should therefore address the above points and corrected them, whenever possible.

Presentation issues:

I am only mentioning the most important ones. There are lots of of bugs, like missing endpoints, missing commas, inconsistent notation, etc. Also, the writing is sometimes too casual. The authors should review the whole document carefully.

1. The notation in Eq (17) is not standard. It is probably better to say that the sequence behaves like the right handside for large $n$. If the authors want to use "lim", it is better to use it for
$Q_n\over {n^{-1-{\gamma\over\sigma}}}$ .

2. In Eq (27) the notation is again confusing (like in Eq (17)), and in this case it clashes directly with (21), where the actual limit is being taken. Again, the authors are recommend to refer to the asymptotic behaviour in the limit of large $t$ in a different way.

3. The digression starting at Eq (39), leading to (47), seems totally unnecessary, since
the final expression (47) is already written in (20), since it obvious that $C_n=\Delta_n(0)$.

4. In section 5.1 the authors are assuming that there is no degeneracy in the values of $\lambda_n$, ie that all $\mu_n$'s are different. This should be clearly stated. (Although is is clear for $\mu_n=\sigma (n+1))

5. The result presented in Eq (66) seems correct. However, the deduction looks confusing.
I suggest that the authors get rid of $\mu[-\partial\over \partial z]$, which is actually never defined, and use instead the actual operator $\sigma d\over dz -1$. The equation (54), which seems correct, is not straightforward, and therefore a greater level of detail would be useful for the reader. It is the authors option, of course, but the arguments leading to (66) are probably clearer if discussed at the level of the differential equation, rather than at the level of the formal solution.

Author Response

Please consider the attached PDF file

(below is the .TEX version)

=================================================

We acknowledge the very detailed report of Reviewer 2. 
We are especially grateful him/her for taking the time to follow 
our calculations in all details and for spotting an important 
issue in one of our derivations.

\vspace{3mm}
We edit our reply to the Reviewer following the structure of the comments in the 
Report:\\

{\bf I. Technical points:}\\

1. The comment about non-positivity of the deviations
is valid and therefore the underlying part has to be corrected. 
Since both $P_n(t)$ and $Q_n$ are normalized 
by their sum to 1, some $\Delta_n(t)$-s must be negative, or all are zero,
since $\sum_n \Delta_n(t)=0$ all the time. 
We agree thus that there is no control over the denominator in 
equations (77) and (81), and -- as a consequence -- the considerations given for 
the monotonic convergence do not hold in the original form.  

Upon the suggestion written in the report,
we corrected the proof of the simple convergence by using inequality 
instead of the equality sign in eqs. (75) and (79).
The conclusions for the convergence remain. 

For the monotonic convergence on the other hand, we have formulated 
a novel compact condition, valid for the simple initial distributions
given in the equations (22) and (23).
For these initial PDF-s with constant $\gamma$ and $\mu$ 
%from the derivative of $\Delta_n(t)$ 
we define now a condition for having monotonic convergence. 

Section 6 has been rewritten throughout. Also, two new graphs are added, 
illustrating the time evolution of $\Delta_n(t)$ for various $n$ values. 

2. In the previous version of the manuscript eq.(28) resulted from eq.(26) 
using $\Gamma[n+1,0]=n!$ and by approximating the exponential pre-factor 
with the linear expression for early times. 
We have indicated this step in the new version of the manuscript. 

3. The forms, suggested by the referee and used by us, are equivalent. 
Note that $\lambda_j/\sigma=j+1+\gamma/\sigma$ therefore  $\lambda_n$ 
exactly replaces the $j=n$ term in the product. 

4. We have checked eq.(74) and its derivation several times, 
by doing independent derivations. They hold.\\

{\bf II. Presentation issues:}\\

1. and 2.: We agree with this comment and we use a simple $\rightarrow$ 
instead of the "$\lim$" notation by now: displaying leading and sub-leading 
terms in the corresponding limit in eq.(17) and eq.(27). Notations are thus modified.

3. The deviations $\Delta_n(t)$ used in equations (39) to (47) were included 
in order to present a second, independent method for obtaining the time evolution.
We agree that it might be an unnecessary detail overloading the main message of 
the article. Thus we moved it into the Appendix section.  

4. It is definitely correct that $\lambda_j$ should have no degeneracy, 
and $\mu_j \ne \mu_k$ in the denominator of the product is obviously assumed. 
In most popular applications to complexity problems a linear preference is taken
in the growth rate, therefore this condition is fulfilled. 
We inserted a note regarding this comment. 

5. Our discussion here is for a general $\mu_n$, not only for the constant 
and linear case, so we tend to keep the $\mu(-d/dz)$ operator. Nevertheless, we 
give now a simple explanation for replacing $\mu_n$, appearing in sums weighted by $P_n(t)$,
by the operator $\mu(-d/dz)$ acting on the generator function. 
This is simply based on  $d/dz \: \exp(-nz) = -n \: exp(-nz)$.  \\

Finally, as suggested by the Reviewers, we revisited style and presentation. \\

We thank again the reviewer for his/her crucial assistance to the improvement of our manuscript.

Round 2

Reviewer 2 Report

See attached file.

Author Response

We acknowledged the main criticism of the Reviewer already from the first Report, but as we detailed in our previous reply letter we considered that it is not appropriate to alter the logic and message of the present manuscript.  We just repeat here our rebuttal:

  1. This work is intended as a contribution for studying in a general manner the transient dynamics of the Growth and Reset model for two specific cases: (a) constant reset and growth rate, (b) linearly increasing growth rate and constant reset rate. There are important applications (which is referenced in the manuscript) and considering here one specific problem would be a totally different approach to the problem. Each application in part would need gathering experimental data on a long time-interval (to have dynamics in the distribution function) and comparison with the dynamical evolution of the experimental density function. Otherwise, one could just have a look on the dynamics presented in Figures 3-5, and without a thorough comparison with experimental data a specific application would not bring any new information. An approach  in such sense would be a completely different study.
  2. The Reviewer ask for including other relevant citations that are not related to our previous work... Unfortunately, up to our knowledge, the whole growth and reset model's history and published applications were done by our team in the recent years, and therefore we cannot name any other relevant work that would deserve citation in the context of the present manuscript. The Reviewer does not suggest anything specifically and encourages to randomly take out some of the really relevant works and put randomly some not too relevant citations. Of course, we do agree that self-citation should not be a method to inflate scientiometric indicators, but if the cited works are indeed relevant for the subject one should not consider this a governing principle in constructing the reference list. The solution suggested by the Reviewer would therefore not help at all in making the present manuscript better.

Reviewer 3 Report

The authors have satisfactorily addressed the main issues raised in my first report.

There remain  however some presentation issues which deserve the authors' attention:

1 - The text of line 41 should appear immediately after Eq (5), which is where $\langle\gamma\rangle$ appears for the first time.

2 - line 42 : "followings" -> "following"

3 - pag 3, beginning of sect 3.2: should stress that $\gamma_n$ are again constant

4 - In the second line below Eq (20): "beats a polynomial" sounds too casual. 

5 - line 61: "obtained" should be replaced by something like "announced", since the result was not obtained in (10-11). 

6 - Eq (23) uses "P_i" whereas $P_n$ is used everywhere else.

7 - It would be better if the authors adhere to just one of the forms $\gamma+\mu$ or $\lambda$.

8 - It would be better if the authors adhere to just one of the expressions "generating function" or "generator function",
preferably "generating function"

9 - The expressions "G space " and "G-space" are not standard (and the authors should adhere to just one of them, if they used it all). It is preferable to use expressions of the type " ... in the generating function formulation .." instead of  " .. in the G-space".

10 - Below eq (36): move "(6)" to after "equation".

11 - Above Eq (40) the authors are using the fact that $c(z)$ turns out to be equal to $\Delta(0,z)$, and not that $c(z)=1$.

Author Response

We are again grateful for the helpful and constructive suggestions of Reviewer 3. All minor comments raised (1-11) in his/her Report were considered exactly as suggested by the Reviewer. Please consider the new version of the manuscript with the modifications as suggested in the Report.